# Antennal Enriched Odorant Binding Proteins Are Required for Odor Communication in *Glossina f. fuscipes*

**DOI:** 10.3390/biom11040541

**Published:** 2021-04-08

**Authors:** Souleymane Diallo, Mohd Shahbaaz, JohnMark O. Makwatta, Jackson M. Muema, Daniel Masiga, Alan Christofells, Merid N. Getahun

**Affiliations:** 1International Centre of Insect Physiology and Ecology (ICIPE), Nairobi P.O. Box 30772-00100, Kenya; jmakwatta@icipe.org (J.O.M.); jackson_mbithi@yahoo.com (J.M.M.); dmasiga@icipe.org (D.M.); 2South African Medical Research Council Bioinformatics Unit, South African National Bioinformatics Institute (SANBI), University of the Western Cape, Private Bag X17, Bellville 7535, South Africa; mohammed@sanbi.ac.za (M.S.); alan@sanbi.ac.za (A.C.)

**Keywords:** *Glossina* sp., odorant-binding proteins, gene expression, structural properties, molecular docking, dsRNAi

## Abstract

Olfaction is orchestrated at different stages and involves various proteins at each step. For example, odorant-binding proteins (OBPs) are soluble proteins found in sensillum lymph that might encounter odorants before reaching the odorant receptors. In tsetse flies, the function of OBPs in olfaction is less understood. Here, we investigated the role of OBPs in *Glossina fuscipes fuscipes* olfaction, the main vector of sleeping sickness, using multidisciplinary approaches. Our tissue expression study demonstrated that GffLush was conserved in legs and antenna in both sexes, whereas GffObp44 and GffObp69 were expressed in the legs but absent in the antenna. GffObp99 was absent in the female antenna but expressed in the male antenna. Short odorant exposure induced a fast alteration in the transcription of OBP genes. Furthermore, we successfully silenced a specific OBP expressed in the antenna via dsRNAi feeding to decipher its function. We found that silencing OBPs that interact with 1-octen-3-ol significantly abolished flies’ attraction to 1-octen-3-ol, a known attractant for tsetse fly. However, OBPs that demonstrated a weak interaction with 1-octen-3-ol did not affect the behavioral response, even though it was successfully silenced. Thus, OBPs’ selective interaction with ligands, their expression in the antenna and their significant impact on behavior when silenced demonstrated their direct involvement in olfaction.

## 1. Introduction

The terrestrial lifestyle of insects has necessitated the adjustment of the olfactory system, such as the evolution of odorant receptors and odorant-binding proteins (OBPs) in flying insects [1,2,3]. The OBPs which have evolved independently from odorant receptor (OR) and earlier functions is not well understood, especially in medically important non-model insects, such as tsetse flies, the biological vector of human and animal trypanosomiasis. The *Glossina* sp. genome expresses several chemosensory proteins including odorant-binding proteins (OBPs) [4,5,6,7]. Human African trypanosomiasis (HAT) is caused by two closely related parasites that are transmitted by tsetse flies [8,9]. In this regards, *Glossina fuscipes fuscipes* (*G. f. fuscipes*) is the most important vector of HAT [10,11,12,13]. More precisely, *G. f. fuscipes* is involved in the transmission of 90% of HAT [14,15]. This tsetse species is also known to have an opportunistic blood-feeding behavior on livestock [16] and therefore contributes to the transmission of African animal trypanosomiasis (AAT). To date, no vaccine is available for HAT or AAT; and vector control tools offer a highly valued approach to disease control. Currently, vector control is achieved through sequential aerial spraying (SAT), ground spraying, insecticide-treated targets or insecticide-treated animals as live baits, the use of traps and the sterile insect technique (SIT) [17,18,19]. In particular, traps and targets have been widely used in tsetse control campaigns in many countries across Africa, despite the fact that type of target and their efficacy largely vary according to the species and the geographical location [20,21,22]. For example, targets 1.0 × 1.0 m in size have been found to be effective for *G. pallidipes* Austen and *G. morsitans morsitans* Westwood [22,23,24,25,26], while small targets 0.25 × 0.25 m in size catch more *G. f. fuscipes* [22,24,25]. It has been suggested that an addition of an appropriate attractant odors such as CO_2_, acetone, 1-octen-3-ol [27,28,29,30,31] and phenols [32,33] could improve the efficacy of the traps; hence enormous effort has been deployed to find efficient combination.

A blend of 3-n-propylphenol, 1-octen-3-ol, p-cresol and acetone enhances trap catches of tsetse flies of the *morsitans* group [21,34]. Moreover, chemicals present in lizard odor can increase the numbers of *G. f. fuscipes* attracted to traps [35]. Despite this enormous effort, we still lack an effective attractants or repellents for many riverine tsetse species, which are both medically and veterinary important vectors. Indeed, the main challenge in finding new or improved tsetse attractants and repellents is that the target olfactory sensory neurons and the chemosensory proteins through which the attractants (and repellents) operate are less investigated. However, recent progress in tsetse genomes has opened a new opportunity to investigate olfactory pathways in *Glossina* sp. [5,6,7].

We believe that a better understanding of tsetse olfaction, the main sensory modality used to locate its hosts, including humans and livestock, will contribute to the improvement of the intervention strategies used to control tsetse fly.

Insect OBPs contain an alpha helical barrel and are characterized by a highly conserved six-cysteine signature (C1-X20-35-C2-X3-C3-X20-30-C4-X8-12-C5-X8-C6) [36,37,38,39,40,41]. The structural characteristics of insect OBPs make them suitable targets for biosensor technology and to identify attractants which could help in the design of environmentally friendly tools for vector control. From the model *Drosophila*, it is well established that OBPs contribute to the sensitivity of the olfactory system [16,17,18,19,20,21,22,23,42]. They elicit a strong binding affinity with odorant compounds in insects like moths. OBPs may also be involved in gain control of the olfactory sensory neuron (OSN) response [43,44,45]. Furthermore, OBPs play a role in social interactions [46], essential amino acid detection [47] for taste and the deactivation kinetics of signal transduction [45]. In insect vectors like mosquitos, OBPs were found to be important for the detection of oviposition attractants [48] and other general odor compounds [48,49,50]. However, the function of OBPs in odor communication in insects in general is still elusive and particularly has not been investigated in non-model insects like tsetse fly. For example, OBP–ligand interactions, their molecular features, tissue-specific expression patterns and their role in odor detection and perception are yet to be studied in *Glossina* sp. Here, we investigated the olfactory function of OBPs using behaviorally well studied odorants. Hence, we aimed in this study to investigate the role of some selected OBPs in *Glossina fuscipes fuscipes*.

To decipher OBPs’ functional roles in olfaction, we first analyzed the structural features of *G. f. fuscipes* odorant-binding proteins. Secondly, tissue-specific expression was done on nine selected OBPs, and then we targeted four OBPs that are expressed more in the antenna (the main olfactory organ) of both males and females to study their olfactory function using RNA interference (RNAi) technology. Our multidisciplinary study demonstrated that OBPs expressed in the antennae are of critical importance for *G. f. fuscipes* olfaction.

## 2. Materials and Methods

### 2.1. Biological Material

All the tsetse flies (*Glossina f. fuscipes*) used in this study were obtained from a colony maintained at the insectary of the International Centre of Insect Physiology and Ecology (icipe). The colony was maintained at 24 ± 1 °C, 75–80% RH (relative humidity) under a 12L:12D photoperiod, and the flies were fed 3 times per week on a silicon membrane with defibrinated bovine blood collected locally [51].

### 2.2. Chemicals

The chemicals (geranyl acetone, δ-octalactone, guaiacol, pentanoic acid and 1-octen-3-ol) were purchased from Sigma-Aldrich (Sigma-Aldrich, Steinheim, Germany) at the highest available purity. For the odorant exposure experiment, chemicals were diluted at 10^−3^
*v*/*v*. Pentanoic acid was diluted in water. Geranyl acetone, δ-octalactone, guaiacol and 1-octen-3-ol were diluted in absolute ethanol (99.8%) [52,53].

### 2.3. Odorant Exposure and RNA Extraction

Odorant exposure was done as previously described by [54,55,56]. Briefly, 25 teneral flies, 2–3 days old, were exposed to different odorant volatiles (geranyl acetone, δ-octalactone, guaiacol, pentanoic acid and 1-octen-3-ol) in a Plexiglas cage measuring 13.5 × 20 × 20 cm for 5 h. The exposure room conditions were similar to the conditions at which the colony was maintained. Males and females were exposed in a separate cage to avoid mating and/or the release of potential sex pheromones, which might interfere with the experiments. After exposure, flies were snap-frozen for 5 min in a −80 °C freezer, and the antennae were harvested on ice. Antennae were manually removed from fly heads and 3 replicates of 150 flies (male–female ratio, 1:1) were used. Dissected antennae of male and female flies were pooled in 2-mL microcentrifuge tubes. To conserve the integrity of the total RNA, the microcentrifuges were deep-frozen in liquid nitrogen during antennae dissection. Tissues were homogenized in a bead mill (TissueLyser LT, Qiagen, Hilden, Germany) for 10 min at 50 Hz and then centrifuged at 13,000× *g* for 5 min, and 350 µL of the homogenate was used for total RNA isolation. TRIzol Reagent (Invitrogen, Thermo Scientific, Waltham, MA, USA 02451) was used to isolate total RNA. RNA quality and quantity were checked with a spectrophotometer (GeneQuant Pro RNA/DNA calculator, Amersham Biosciences, Cambridge, UK).

### 2.4. Quantitative Real-Time RT-PCR Assay and Data Analysis

A High Capacity cDNA Reverse Transcription kit (Applied Biosystems, Foster City, CA, USA) was used for cDNA synthesis according to the manufacturer’s instructions. For this, 500 ng of total RNA was reverse transcribed in a reaction mixture with a final volume of 20 μL. The cDNA was amplified with 5× HOT FIREPol EvaGreenqPCR Mix Plus (ROX) (Solis BioDyne Inc., Teaduspargi, Estonia) according to the manufacturer’s instructions. Real-time PCR was carried with QuantStudio 3 (Applied Biosystems 7500, USA) using the comparative delta-delta CT (ΔΔCT) method as previously described [57]. The primer sets (Appendix A) were designed with Primer3 software (http://bioinfo.ut.ee/primer3-0.4.0/ accessed on 30 June 2019) [58] and optimized by gradient PCR (using a Kyratec Thermal cycler). Additionally, tissue-specific expression was done using antennae and legs using real-time quantitative PCR. The legs were chosen because several studies have demonstrated the abundance of OBPs in that part of the body [59,60,61,62]. The tissue expression was used to validate the quality of the primers and also to identify a potentialodor binding protein. The PCR products were loaded on 1.5% ethidium bromide-stained agarose gel and visualized using ultraviolet light.

### 2.5. Structural Analysis of G. f. fuscipes Odorant-Binding Proteins

Sequences for *G. f. fuscipes* OBPs were retrieved from VectorBase (www.vectobase.org) (accessed on 10 June 2019) and compared by multiple alignments performed by Multiple Sequence Comparison by Log-Expectation (MUSCLE) after removal of signal peptides. The resulting alignment was viewed and manually edited using Jalview [63]. The signal peptide screening was performed using the SignalP-5.0 webserver (http:/www.cbs.dtu.dk/services/SignalP) (accessed on 10 August 2019) [64]. SignalP was chosen because of its reliability compared with the other available tools [64,65,66]. It combines deep learning and recurrent neural networks to predict signal peptides [64].

### 2.6. Homology Modeling and Binding Pocket Analysis

Homology modeling was done using he Protein Homology/analogy Recognition Engine V 2.0 (Phyre2) [67]. The intensive mode and *ab initio* techniques were used to perform complete modeling of the entire proteins. Swissmodel and I-tasser were used to generate a comparative model but Phyre2 offered better models. The quality of our predicted models was checked using SAVES v5.0 (https://servicesn.mbi.ucla.edu/SAVES) (accessed on 10 August 2019) tools, ProSA and Qmean. The binding pocket was identified using the DoGSite Scorer (http://proteins.plus) (accessed on 10 August 2019) [68]. The pocket’s size, shape and functional descriptors were compared and analyzed.

### 2.7. Model Optimization and Molecular Docking

The models were optimized and docked using ICM-Pro version 3.5 (Molsoft LLC, San Diego, CA, USA). Five docking scores were generated, which corresponded to five different conformations. The best docking score for each odorant-binding protein was used for statistical analysis and to build the heatmap graph. The heatmap graph was generated in R version 3.5.1 [69]. Ligplot+ was used to generate the 2D interaction diagram of different complexes.

### 2.8. Molecular Dynamic Simulations

The 4 top-scoring docked complexes were selected and subjected to Molecular dynamic (MD) simulations using the GROMACS 2018-2 software package [70]. Initially, the topologies of the protein structures in the docked complexes were generated using the GROningen MOlecular Simulation (GROMOS96) 53a6 force field [71], and the PRODRG server was used for parameterization of the complex ligand [72]. The latter server does not contain the functional module for calculation of the partial charges. Consequently, the Density Functional Theory (DFT) method implemented in GAUSSIAN with the B3LYP 6-31G (d,p) basis set in combination with the CHarges from ELectrostatic Potentials using a Grid (CHELPG program) [73] was used for charge correction. After parameterization, the solvation of docked complexes was performed using a SPC/E water model [74], which was followed by neutralization, in which countering sodium (Na) and chlorine (Cl) were added to stabilize the systems. As a result, the solvated and neutralized systems were energetically minimized in the consecutive step using the combined steepest descent and conjugate gradient algorithms, with a convergence criterion of 0.005 kcal/mol. Afterwards, the position restraints were applied to the structures of system ligands before the equilibration phase. 

The equilibration step was carried out in combined constant volume NVT (with N = moles, V = volume and T = temperature) and constant pressure NPT (with N = moles, P = Pressure and T = temperature ensemble stages, each at a 100 ps timescale. The temperature of 300 K was maintained for the system using the Berendsen weak coupling method, and pressure of 1 bar was maintained utilizing a Parrinello–Rahman barostat in the equilibration stage. In the final production stage, the conformations were generated using the LINear Constraint Solver (LINCS) algorithm for a 100 ns timescale, and trajectories were generated, which were analyzed to understand the behavior of each complex in the explicit water environment. The changes in the H-bonds, protein–ligand distances and radius of gyration (Rg), as well as the root mean square deviations (RMSD), were analyzed using the GROMACS utilities. Furthermore, the *g_mmpbsa* package was used for calculation of the free energies of interactions between the complexed protein–ligand systems using the principles of Molecular Mechanics Poisson–Boltzmann Surface Area (MM-PBSA) protocols [75].

### 2.9. dsRNAi Preparation and Its Delivery to Flies

dsRNAi targeting 4 OBP genes (GffObp19a, GffObp83a1, GffObp83a2 and GffObp83a4) was prepared from PCR amplicons tailed with T7 promoter sequences using the Replicator RNAi kit (Finnzymes) according to the manufacturer’s instructions. The PCR amplification was done using gene-specific primers which were manually designed from the coding sequence (CDS) of each gene (Appendix A). To confirm the specificity of the primers, the PCR product was sent to Macrogen Europe B.V. Amsterdam, the Netherlands, for sequencing. For the PCR template, we used cDNA synthesized from total RNA using a High Capacity cDNA Reverse Transcription kit (Applied Biosystems, Foster City, CA, USA). The total RNA was extracted from the antennae and legs of 150 flies (male–female ratio, 1:1). TRIzol Reagent (Invitrogen, Thermo Scientific) was used to isolate total RNA, and RNA quality and quantity were assessed with a spectrophotometer (GeneQuant Pro RNA/DNA calculator, Amersham Biosciences, Cambridge, UK). Contaminating genomic DNA was removed from the transcription reaction by DNase treatment. dsRNAi was eluted in nuclease-free water and the concentrations were measured using a spectrophotometer (GeneQuant Pro RNA/DNA calculator, Amersham Biosciences, Cambridge, UK).

dsRNAi was delivered to flies by feeding it through pre-warmed (at 37 °C) bloodmeal containing the dsRNA (Figure 4A,B). The protocol of mixing the bloodmeal and the dsRNAi was adapted from [76]. Twenty teneral flies were fed in a cage and kept under insectarium conditions for 72 h after dsRNAi feeding. Approximately 10 µL of dsRNAi diluted to appropriate concentrations in nuclease-free water was added to 500 µL of bloodmeal. Unfed flies were automatically removed after feeding. For each experimental group, 80 flies were used: 20 flies were used for the gene silencing efficiency check at 96 h post-feeding and 60 flies were used for the behavioral assay. Nuclease-free water was used as the internal control.

### 2.10. Behavioural Bioassay with dsRNAi Gene-Silenced Flies

To assess how the silencing of different OBPs affected the behavior of the flies, an attraction bioassay was performed in a plastic cage (length = 75 cm, width = 30 cm, height = 45 cm) containing two sticky papers (13 × 10 cm) used as a trap (Figure 6D). The bioassay was done in the same conditions as in the insectary with 24 ± 1 °C, and 75–80% RH (relative humidity) under a 12L:12D photoperiod. On the sticky paper, we placed a cotton roll (100 mm × 15 mm), which served as a dispenser. For an attractive odor source, 100 µL of 10-3 *v*/*v* 1-octen-3-ol diluted in mineral oil was loaded on the cotton roll dispenser, with 100 µL of mineral oil only to serve as a control. In the cage, 20 flies (male and female in a 1:1 sex ratio) were starved for 3 days and introduced, and each experiment was replicated 3 times. The flies were introduced into the cage 20 min before loading the attractive odor and the mineral oil. Flies were allowed to choose between the attractive source and the control for 24 h; afterwards, the attraction was scored and the attraction index (AI) was calculated using:AI = (N_odour_ − N_control_)/N_total_(1)
where N_odour_ corresponds to the number of flies trapped at the odor source, N_control_ is the number of flies trapped at the control and N_total_ is the number of flies used for the assay [77]. The significant differences between the attraction indices were noted using the analysis of variance (ANOVA) test followed by Tukey’s HSD (Honestly Significant Difference) post hoc test, owing to the normality of the data (Shapiro test: *p* > 0.05) and the homogeneity of the variance (Levene test: *p* > 0.05).

## 3. Results

### 3.1. Structural Analysis of Glossina f. fuscipes Odorant-Binding Proteins

To study the function of *G. f. fuscipes* OBPs, we retrieved all the putative OBPs that have been previously identified in *G. f. fuscipes* from VectoBase [6]. In total, 23 odorant-binding proteins were analyzed for their molecular structural features. The molecular weight of these OBPs ranged between 12 and 30 kDa, the predicted OBP sequences were encoded by between 107 and 258 amino acids with a low sequence similarity. The signal peptides screening showed three OBPs (GffObp44a, GffObp57c and GffObp84a) lack a signal peptide sequence, but the other 19 OBPs had a signal peptide at the *N*-termini. According to the number of conserved cysteine motifs, odorant-binding proteins (OBPs) are divided into four groups (classic OBPs, minus-C OBPs, plus-C OBPs and atypical OBPs). Our structural analysis of G. *f. fuscipes* OBPs clustered in to three different classes: the classic OBPs, the minus-C OBPs and the atypical OBPs, as described in previous studies [4,78]. Twelve OBPs (GffObp19, GffObp19b, GffObp28a, Gffobp56d, Gffobp56e, Gffobp57c, Gffobp69a, Gffobp83a4, Gffobp83g, Gffobp84a, Gffobp99d and GffLush), showed six conserved cysteine (C1–C6) motifs and hence were classified as classic OBPs. GffObp44a, GffObp8a and Gffobp99b showed fewer than six conserved cysteine motifs and were classified as minus-C OBPs. Several OBPs (Gffobp19a, GffObp19c, GffObp56h, GffObp56i, GffObp83a1, GffObp83a2, GffObp83cd and GffObp83ef) had more than six cysteine motifs and were classified as atypical OBPs (Figure 1). We did not find any plus-C OBPs that had a proline residue next to the sixth conserved cysteine in the *G. f. fuscipes* OBPs analyzed. 

### 3.2. In Silico Homology Modeling and Binding Pocket Analysis of G. f. fuscipes OBPs 

To investigate the interactions between OBPs and our selected odorant compounds, we conducted molecular docking to predict the binding affinity and to further select the best OBPs for functional analysis using RNAi silencing-based techniques. 

The homology modeling was done using the Phyre2 web server. We found nine (9) models with a template similarity of <30% (Appendix A). The quality assessments of these 3D models were further evaluated and validated for the molecular docking (Appendix A).

GffObp44a and Gffobp83a4 exhibited the smallest pocket size and volume (Appendix A). Traditionally, OBPs contain six α-helices and three disulfide bridges with an internal binding cavity. All the analyzed OBPs showed similar structural characteristics, except GffObp44a (which had five α-helices) and GffObp69a (seven α-helices) (Appendix A). Their binding pocket did not possess any clear binding cavity on the surface and no subpocket was found in these OBPs (Figure 2). The biggest binding pockets were observed in GffLush, GffObp19a, GffObp69a, GffObp83a1, GffObp83a2, GffObp83g and GffObp99d (Figure 2), whereas GffObp83g had a relatively small volume. GffLush, GffObp19a, GffObp83a1, GffObp83a2 and GffObp99d presented more than one binding cavity, while GffOb69a and Gff83g had a unique and clear binding cavity on the surface. 

### 3.3. Tissue-Specific Expression of Different OBPs 

To assess the potential olfactory function of the selected OBPs, we performed a tissue-specific expression analysis in the antennal tissues and different legs (front and hind) of male and female *G. f. fuscipes* using quantitative real-time polymerase chain reaction (PCR). Except for GffObp44a and GffObp69, all the studied OBPs were expressed in the antenna. However, GffObp83a1, GffObp83a2, GffObp83a4, GffObp19a and GffLush were highly enriched in the antennae. GffObp19a had 50× expression in the antennae compared with the legs, and GffObp83a2 was 100× expressed in the antennae. GffObp83a1 and GffOpb83a4 were highly expressed in all the tissues. GffObp44a and GffObp69a were not expressed in the antennae but detected in the legs (Figure 3). GffObp99d was expressed in female and male legs, as well as in the antennae. GffLush was expressed in both the antennae and legs in both males and females.

### 3.4. The Olfactory Function of Glossina f. fuscipes OBPs Expressed in the Antennae 

To investigate the olfactory role of OBPs that are more expressed in the antennae in odor communication, we conducted behavioral response assays by comparing wild-type flies’ responses against flies where OBPs were individually silenced. Herein, we evaluated the potential role of four OBPs (GffObp19a, GffObp83a1, GffObp83a2 and GffObp83a4) that are expressed in the antennae of both males and females. We silenced these OBPs using an dsRNAi interference technique and evaluated the behavioral impact using a free flight bioassay (Figure 4D). Our result showed that efficient silencing of OBPs can be achieved within 96 h when flies were offered bloodmeal containing the dsRNAi of specific OBPs (Figure 4A–C). Furthermore, OBP gene silencing efficiency varied between OBPs; for example, the silencing of Gffobp83a2 was minimal compared with the other three OBPs (Figure 4C). The mortality of flies fed with dsRNAi was minimal (1/20 flies), which was the same as in the control flies. 

During the behavioral assay, the dsRNAi-fed flies were flying normally compared with the control flies, showing that OBPs silencing did not affect their flight ability. For the behavioral assay, the silencing of *Glossina f. fuscipes* Obp19a did not affect the attraction of *G. f. fuscipes* to 1-octen-3-ol as compared with the wild-type (*p* = 0.73); (Figure 4E, Appendix A). The attraction index (AI) was 0.55, which meant that the flies were attracted. However, the silencing of *Glossina f. fuscipes* GffObp83a1, GffObp83a2 and GffObp83a4 (Figure 4C) significantly reduced the flies’ attraction to 1-octen-3-ol as compared with the wild-type and nuclease-free water flies, (*p* = 0.008 for Obpa83a1, *p* = 0.001 for GffObp83a2 and *p* = 0.003 for Obp83a4) (Figure 4E).However, the negative control flies fed on nuclease-free water had a similar attraction index as the wild-type (*p* > 0.05) and effect on the expression of OBPs (Figure 4B,E, Appendix A). 

### 3.5. Physiochemical Properties and Molecular Docking of G. f. fuscipes OBPs

To understand the dynamics of binding affinities of the OBPs, we analyzed their physicochemical properties such as hydrogen bond donors or acceptors and the number of hydrophobic interactions present in the binding pockets. All the analyzed OBPs possessed more H bond acceptors than H bond donors (Appendix A). GffLush, GffObp83a4, GffObp83a1, GffObp83a2 and GffObp44a had the smallest number of H bond donors (13 or fewer). We noted a high number of H bond donors in GffObp83g, GffObp99d, GffObp19a and GffObp69a (Appendix A). The binding pockets with a high number of hydrophobic interactions were observed in GffObp19a, GffObp83a2, GffObp99d, GffObp69a and GffLush.

Having analyzed the physicochemical properties of the OBPs, we then performed a molecular docking using Waterbuck Repellent Blend (WRB) compounds (pentanoic acid, δ-octalactone, geranyl acetone, and guaiacol), which inhibited the blood-feeding behavior of *G. f. fuscipes*, and 1-octen-3-ol, which enhanced blood-feeding [56]. Using the lowest docking score, we conducted unsupervised hierarchical clustering to identify ligand–OBP interaction patterns (Appendix A). Three clusters were observed: the first cluster included Gffobp83g and GffObp69a. These OBPs strongly interacted with pentanoic acid, δ-octalactone, geranyl acetone and guaiacol (Appendix A).

The second cluster consisted of GffObp19a, GffLush and GffObp44a, highlighting their strong affinities to geranyl acetone, guaiacol and 1-octen-3-ol. Lastly, GffObp83a1, GffObp83a2, GffObp83a4 and GffObp99d were clustered together. Their best docking scores were observed with δ-octalactone, geranyl acetone, guaiacol and 1-octen-3-ol, whereas GffObp83a4 and GffObp99d also interacted with pentanoic acid (Appendix A). 

### 3.6. Conformational Dynamics of Docked Systems

To assess the efficiency of our docking and to understand the binding and its mechanisms, we conducted a molecular dynamics (MD) simulation using the functionally characterized OBPs and 1-octen-3-ol, the odor we used for our behavioral assay. 

The structural attributes of the docked systems were explored using the principles of MD simulations at the timescale of 100 ns [56]. The structural compactness and stability of the docked systems were analyzed in terms of the calculated radius of gyration (Rg) and the root mean square deviation (RMSD) values (Appendix A). The variations in the Rg values for all the systems were reported, and it was observed that the GffObp19a–1-octen-3-ol system showed the highest level of structural compactness in which the Rg values were fluctuating between 1.25 nm and 1.3 nm (Appendix A), while for the rest of the system, the values were observed to be higher than 1.3 nm, highlighting the attainment of less compactness. Similarly, the RMSD value projections showed that the GffObp19a–1-octen-3-ol system achieved the least stability among the studied systems, for which the values fluctuated between 0.4 nm and 0.5 nm. All the other three systems showed relatively similar stability profiles in which variations in the RMSD values were observed between 0.2 nm and 0.4 nm (Appendix A). 

The closeness between the proteins and the docked ligands in the studied systems was understood in terms of the calculated distances between the interacting molecules. The GffObp83a4–1-octen-3-ol system, followed by the GffObp83a–1-octen-3-ol system, showed the shortest distances in the respective systems, which indicated that a higher degree of interaction was observed in the respective systems as compared with the rest, with the lowest calculated distances observed in the GffObp19a–1-octen-3-ol system (Appendix A). 

The hydrogen bond (H bond) patterns were further explored for understanding the nature of the interactions between the proteins and docked ligands during MD simulations. The GffObp19a–1-octen-3-ol system showed the presence of up to four H bonds, while around three H bonds were observed in the GffObp83a1–octen-3-ol and GffObp83a4–1-octen-3-ol docked systems. The GffObp83a2–1-octen-3-ol system showed the presence of only two H bonds, which indicated that a comparatively lower interaction level was observed in the respective system (Figure 5). 

Furthermore, MM-PBSA based protocols were used for calculation of the free energies of the interactions between the interacting molecules of the docked systems (Figure 6). The GffObp83a–1-octen-3-ol, GffObp83a2–1-octen-3-ol and GffObp83a4–1-octen-3-ol systems showed relatively similar patterns of the total interaction energy, in which the values were observed to be between −100 kJ/mol and −150 kJ/mol, while the GffObp19a–1-octen-3-ol system showed a relatively lower binding affinity, which can be deduced from the calculated total free energy of binding, calculated to be between −50 kJ/mol and −100 kJ/mol. 

### 3.7. Evaluation of Deorphanization of Receptors Based on Expression Alterations in mRNA Levels (DREAM) on Different OBPs 

To assess the role of selected odorant-binding proteins in olfaction, we analyzed the gene expression alteration patterns in the antennae after exposure to different odorant compounds. The gene expression analysis showed that the mRNA transcript levels of GffObp19a, GffObp83a1, GffObp83a2, GffOb83a4 and GffObp99d were upregulated in the antennae when the *G. f. fuscipes* were exposed to δ-octalactone. Furthermore, the mRNA transcript levels of GffLush and GffObp44a were not affected when the flies were exposed to δ-octalactone (Figure 7A). For geranyl acetone exposure, GffObp19a, GffObp83a4 and GffObp83g were upregulated; GffObp83a1, GffObp83a2 and GffObp99d were downregulated (Figure 7B). Meanwhile, GffObp44a and GffLush did not show any change in mRNA transcript levels. The exposure to guaiacol affected four odorant-binding proteins: GffObp83a1 and GffObp83a2 were downregulated, while GffObp83a4 and GffObp99d were upregulated (Figure 7C). The mRNA transcript levels of GffObp83g, GffObp69a and GffObp83a4 did not change upon exposure to pentanoic acid. On the other hand, GffObp83a1 and GffObp99d were upregulated, while GffObp44a, GffObp83a2 and GffLush were downregulated (Figure 7D).

The exposure to 1-octen-3-ol did not affect GffLush, GffObp44a and GffObp69a. However, GffObp19a, GffObp83a1, GffObp83a4 and GffObp99d were significantly upregulated, while GffObp83g and GffObp83a2 were significantly downregulated when exposed to 1-octen-3-ol (Figure 7E).

## 4. Discussion

In this study, by using multiple approaches (i.e., tissue expression, structural, ligand interaction, molecular dynamics, silencing and behaviour), we demonstrated the essential olfactory function of OBPs expressed in the antennae of tsetse flies. Our results show that *Glossina f. fuscipes* odorant-binding proteins are subdivided into three subfamilies (minus-C, classical and atypical OBPs) [2,36,41,79,80]. Minus-C OBPs present an intermediate structure in the functional evolution of OBPs [81]. In the present study, three minus-C OBPs were found: GffObp44a, GffObp8a and Gffobp99b in *G. f. fuscipes*. GffObp44a, which was expressed only in the female hindlegs, is an OBP without signal peptides (SPs) and showed a small binding pocket with a smaller number of hydrophobic interactions compared with the OBPs with signal peptides. The function of such OBPs need further investigation [78]. 

OBPs without SPs have also been identified in other insects [78]; however, the role of signal peptides in the interactions between OBPs and ligands remains unclear. The physicochemical properties of these OBPs from the structural analysis suggest that they could be important for the general odorant-binding proteins (GOBPs), as they contribute to rendering the binding pocket more hydrophobic, thus allowing higher flexibility of the pocket towards general odorants [82]. However, OBPs without signal peptides are considered to be mature proteins and their binding function could be limited to small ligands because of the shape and dynamics of their binding pocket. Otherwise, signal peptides were also found to play a role in protein stability [83]; more recently, it was suggested that the signal peptide at the *N*-terminal end could be used for designing highly specific primers and probes to detect the expression patterns of odorant-binding protein genes in the main olfactory and gustatory organs [84]. 

Classical and atypical OBPs have been extensively studied in different insects and considered as key players in olfactory processing [37,39,44,85,86,87,88]. Similarly, classical and atypical OBPs showed binding pockets that are suitable for binding diverse odorants. This was supported by our mRNA transcriptome alteration, which is a proxy of ligand–OBP interaction results, whereby the given classical and atypical OBPs interacted with more odorants, and a given odorant activated more classical and atypical OBPs. It is well established that the dynamics of protein binding pockets are crucial for their interaction efficiency and specificity [82,89,90,91]. The shape and the volume of the classical (739 Å^3^ to 1389 Å^3^) and atypical (935 Å^3^ to 1429 Å^3^) binding pocket cavities and their structural flexibility allowed us to postulate that they are suitable for various ligand binding. This is also supported by the physicochemical properties of the binding cavities, where we observed several hydrophobic interactions and hydrogen bonding in the binding cavities, as previously reported for other insects [92,93]. Hydrophobic interactions considerably reduce the undesirable interactions with water molecules, thus increasing the efficiency of receptor–ligand interactions. The molecular docking results showed that pentanoic acid interacted with GffLush, GffObp44a, GffObp83a1 and GffObp83a2, while δ-octalactone, geranyl acetone, guiaicaol and 1-octen-3-ol interacted with Gffobp19a, GffObp83a1, GffObp83a2 and GffObp99d. The binding pockets of these four OBPs were also found to have a large volume and area, and better physicochemical characteristics such as hydrogen bonding and hydrophobic interactions. These observations are in line with previous studies [90,93,94] on the correlation between binding pocket dynamics and the flexibility of the proteins to adapt their binding affinity to different molecules The structural basis for this flexible chemical recognition remains unknown. A given OBP interacted with more than one odorant with diverse chemistry. For example, in odorant receptors (ORs), a recent study [95] showed that odor binding is mediated by hydrophobic non-directional interactions with residues distributed throughout the binding pocket on the ORs, enabling the flexible recognition of structurally distinct odorants. Similarly, these OBPS have a highly hydrophobic interaction (Appendix A). GffObp19a, GffObp83a1 and GffObp83a2, which are expressed in both male and female antennae, presented the best physicochemical features, such as hydrogen acceptors/donors and hydrophobic interactions, and they interacted efficiently with ligands (Figure 7). We further investigated the function of GffObp19a, GffObp83a1, GffObp83a2 and GffObp83a4 by RNAi-based silencing. The studied OBPs showed variations in their expression between tissues (antennae and legs) and sexual dimorphism. For instance, except for GffObp69a and GffObp44a, all the other OBPs were expressed in the antennae, whereas GffObp69a was only expressed in the female hind legs. This finding is neither unique nor surprising, as there are sex-specific ecological and physiological behaviors. In *Drosophila*, DmObp69a has been shown to be involved in social interactions [46] and is probably involved in the detection of contact sex pheromones. The selective expression of GffObp69a and Gffobp44a in the female leg is not clear but might be associated with female-specific behavior such as larviposition, which needs further investigation. Almost all the other OBPs are expressed in *G. f. fuscipes* legs, indicating that they might also have a role in social interactions such as sexual behavior [96]. 

We found that, like ORs, OBP mRNA expression levels were altered by up- and downregulation and others were not affected when the flies were exposed to WRB and 1-octen-3-ol, showing that OBPs are also selective, the same as odorant receptors [56]. GffLush, which was conserved between sexes in the tissues in its expression, is considered to be a pheromone-binding protein in *Drosophila melanogaster* and did not exhibit a strong interaction with any of the WRB components and 1-octen-3-ol. Four antennal-enriched OBPs were selected for their functional study using RNAi-mediated gene silencing. RNAi silencing via dsRNAi feeding indicated that it is possible to silence OBPs in non-model insects such as tsetse flies and investigate their function. The target gene interference was efficient, which indicates the effectiveness of studying insect OBPs using dsRNAi silencing [63,76,83]. We successfully silenced four OBPs expressed in the antennae of both sexes in *Glossina f. fuscipes*, which have demonstrated to favor physiochemical properties and, three of the four OBPs, reduced *G. f. fuscipes* flies’ behavioral response to 1-octen-3-ol as compared with control and wild-type flies. Our behavioral assay demonstrated that GffObp83a1, GffObp83a2 and Gff83a4 play an important role in the detection and perception of 1-octen-3-ol. Their silencing significantly reduced the attraction of the flies to 1-octen-3-ol, which is a known attractant of tsetse fly. GffObp19a was found to have less effect on the perception of 1-octen-3-ol. 

To understand the dynamics of the binding processes of 1-octen-3-ol to the four OBPs, we explored the molecular dynamics and ligand–OBP interaction patterns. We found that the gene silencing results were in line with the in silico predictions of the interactions of GffObp83a1, GffObp83a2 and Gff83a4 with 1-octen-3-ol. The molecular dynamics showed similar patterns. The lack of GffObp19a silencing in response to 1-octen-3-ol supported by our molecular dynamics studies that did not elicit good stability during the molecular dynamics simulation compared with the complexes formed by GffObp83a1, GffObp83a2 and GffObp83a4. Few hydrogen bonds were observed in the molecular dynamics simulation, while hydrophobic interactions (Van der Waals) were elicited in the 2D interaction diagram (Figure 6). Furthermore, the change in mRNA expression in GffObp19a when exposed to 1-octen-3-ol might be a false positive effect from the experiment [55].

In the binding cavities, GffObp83a1 (Tyr145, Phe146 and His144), GffObp83a2 (Tryp140, Phe149 and Tyr148) and Gff83a4 (Try139, His146, Tyr147 and Phe148), which affected the behavioral response, showed good hydrophobic interactions compared with GffObp19a (Appendix A). Similar binding patterns were observed by [93], where they found hydrogen bonding to be less important than hydrophobic interactions. The reduced response of OBP-silenced flies supports the hypothesis that OBPs are important for odor communication. Similarly, reducing the expression of DmelOBP59a affected the detection of attractant odorants in *Drosophila* [42,97]. 

Cumulatively, our study shows clear evidence of the role of GffObp83a1, GffObp83a2 and GffObp83a4 in the detection and perception of 1-octen-3-ol by *G. f. fuscipes*. Similarly, the reduced expression of DmelOBP59a in *Drosophila* affects the detection of 1-hexanol, 2-heptanone, and propanal, and a decrease in bitter taste consumption [42,97]. We successfully silenced OBPs, which enabled us to study the function of certain OBPs in non-model insects like tsetse flies using a dsRNAi feeding approach. However, it could be interesting to study how long the dsRNAi can stay stable in the blood.

In summary, the olfactory tissue expression, the selective mRNA alterations when exposed to odorants and their significant effects when silenced on the behavioral response demonstrate that OBPs are directly involved in odorant detection and perception. Furthermore, these OBPs vary in their physiochemical structures, which might affect their ligand interaction and selectivity, and their various potential roles in olfactory function. Furthermore, the sexual dimorphism and tissue-specific expression indicate their involvement in various sensory modalities such as olfaction, including sexual interaction, and taste. We believe that a better understanding of OBPs in chemical communication will contribute to the more efficient development of olfactory-based tools, such as sensors, as well as control tools such as attractants and repellents for tsetse fly, a vector of sleeping sickness and nagana. 

## Figures and Tables

**Figure 1 biomolecules-11-00541-f001:**
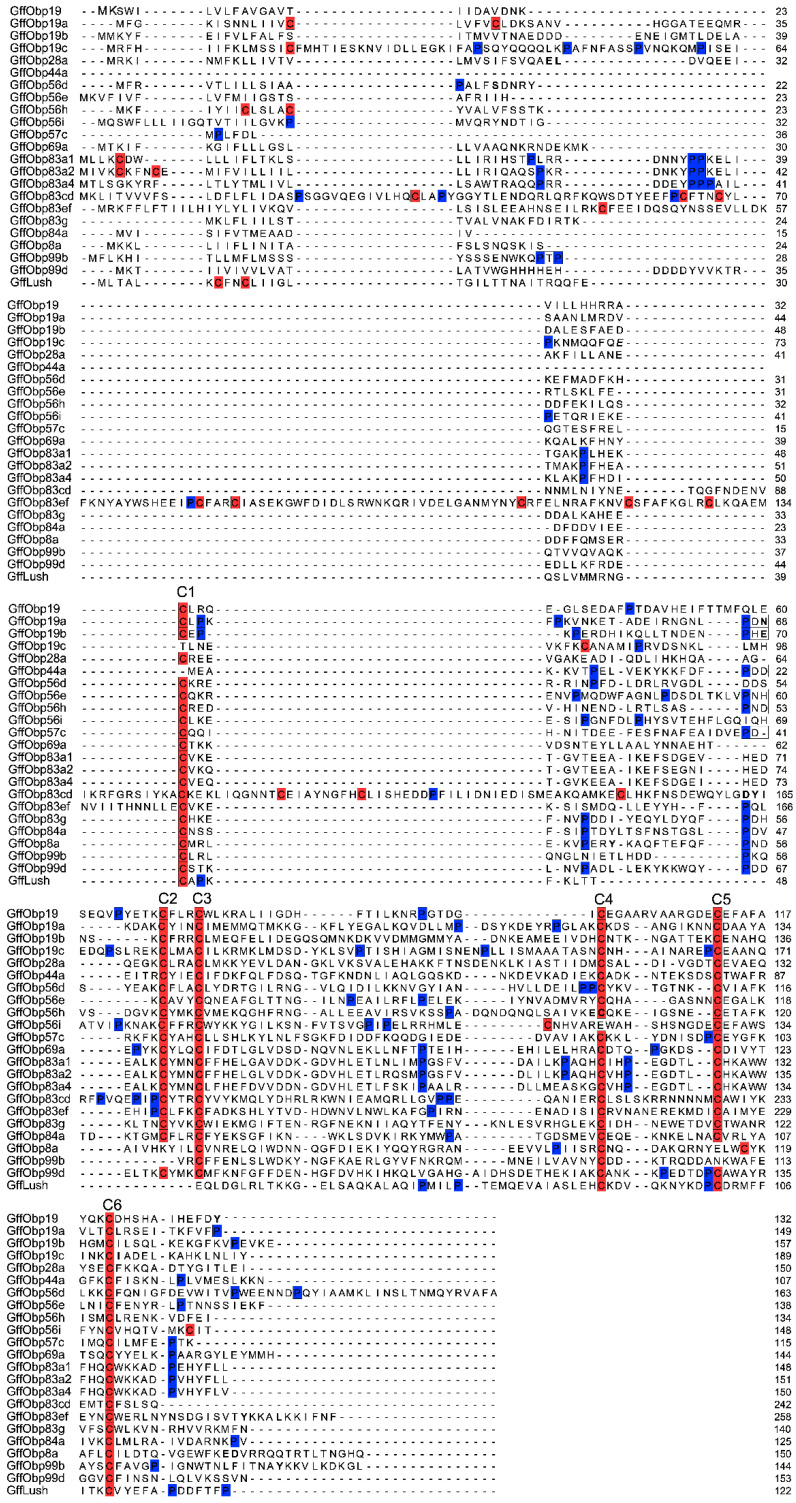
Multiple alignments of *G. fuscipes fuscipes* odorant-binding protein genes. Conserved cysteines are highlighted in red and proline residues in blue.

**Figure 2 biomolecules-11-00541-f002:**
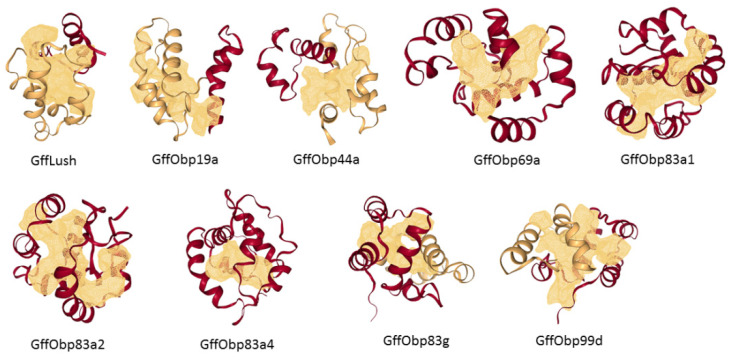
Structural features of nine odorant-binding proteins of *Glossina f. fuscipes.* 3D structure showing the α-helices. Gold color shows the protein surface topology, highlighting the binding pocket.

**Figure 3 biomolecules-11-00541-f003:**
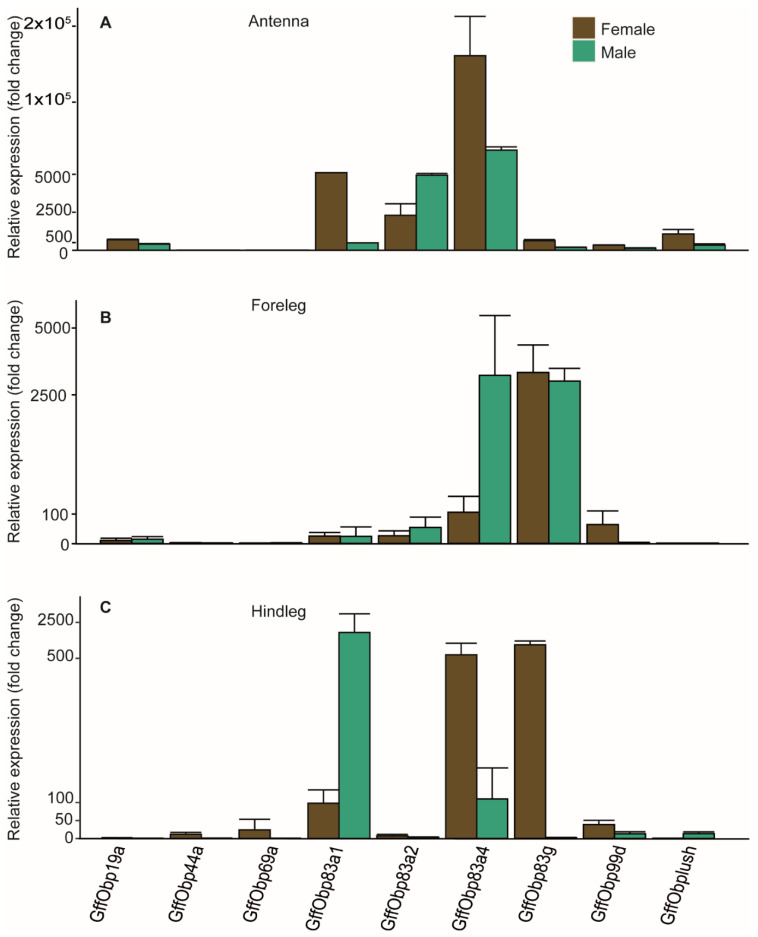
Tissue and sex-specific expression of odorant-binding proteins in antennae and legs using RT-qPCR. B-actin was used as an internal control to normalize the data, and the 2−ΔΔCT method [79] was used to calculate the expression level. Bars represent the ± standard error. Given the big variation of expression in different tissues, the graphs are not presented at the same scale. (**A**) Tissue and sex-specific expression of odorant-binding proteins in Antenna (**B**) Tissue and sex-specific expression of odorant-binding proteins in forelegs (**C**) Tissue and sex-specific expression of odorant-binding proteins in in hindlegs.

**Figure 4 biomolecules-11-00541-f004:**
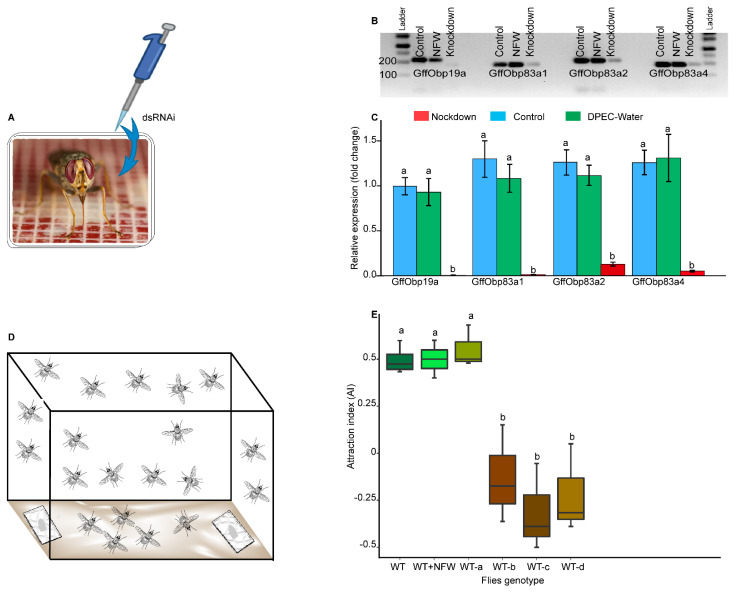
Schematic representation of the RNAi experiment and its effect on odorant-binding protein (OBP) expression and on fly behavior. (**A**): dsRNAi delivery and fly conditioning after dsRNAi intake. (**B**,**C**): comparative expression of different OBPs in wild-type and knockdown flies using real-time qPCR. (**D**): behavioral assay set-up. (**E**): Boxplots illustrating the attraction index (AI) of various *G. f. fuscipes* genotypes (WT = wild-type, WT + nfw = nuclease-free water, WT-a = Obp19a-silenced, WT-b = Obp83a1-silenced, WT-c = Obp83a2-silenced= and WT-d = Obp83a4-silenced. Box plots with different letters are significantly different from each other using ANOVA followed by Tukey’s test.

**Figure 5 biomolecules-11-00541-f005:**
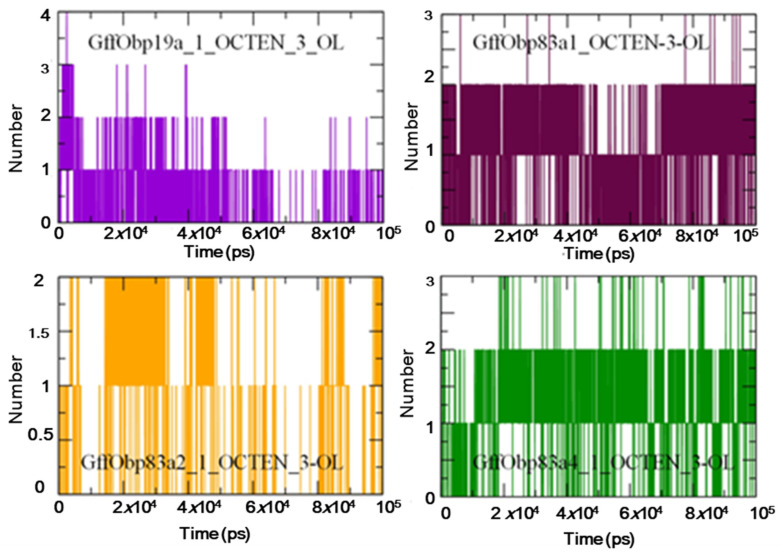
Graphs illustrating the variations observed in the pattern of hydrogen bonding between the studied proteins and ligands during 100 ns molecular dynamics (MD) simulations.

**Figure 6 biomolecules-11-00541-f006:**
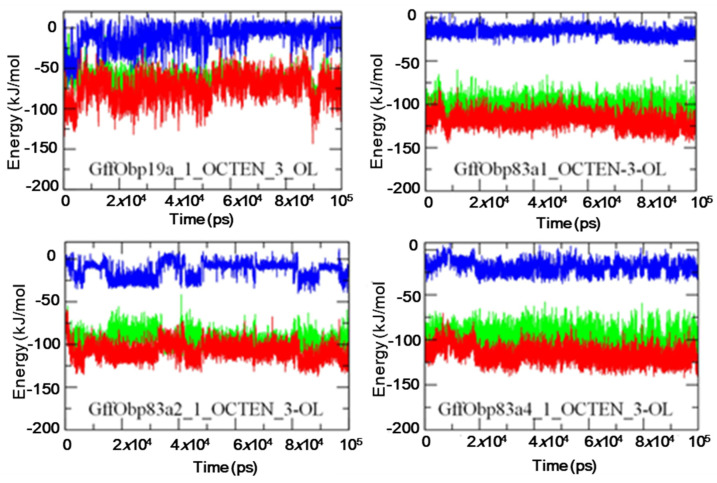
The Molecular Mechanics Poisson–Boltzmann Surface Area (MM-PBSA)-based calculated energy curves showing the variations in the interaction energies observed between the proteins and docked ligands during 100 ns molecular dynamics (MD) simulations. (light green, Vander Waals energy; blue, electrostatic energy; red, total energy).

**Figure 7 biomolecules-11-00541-f007:**
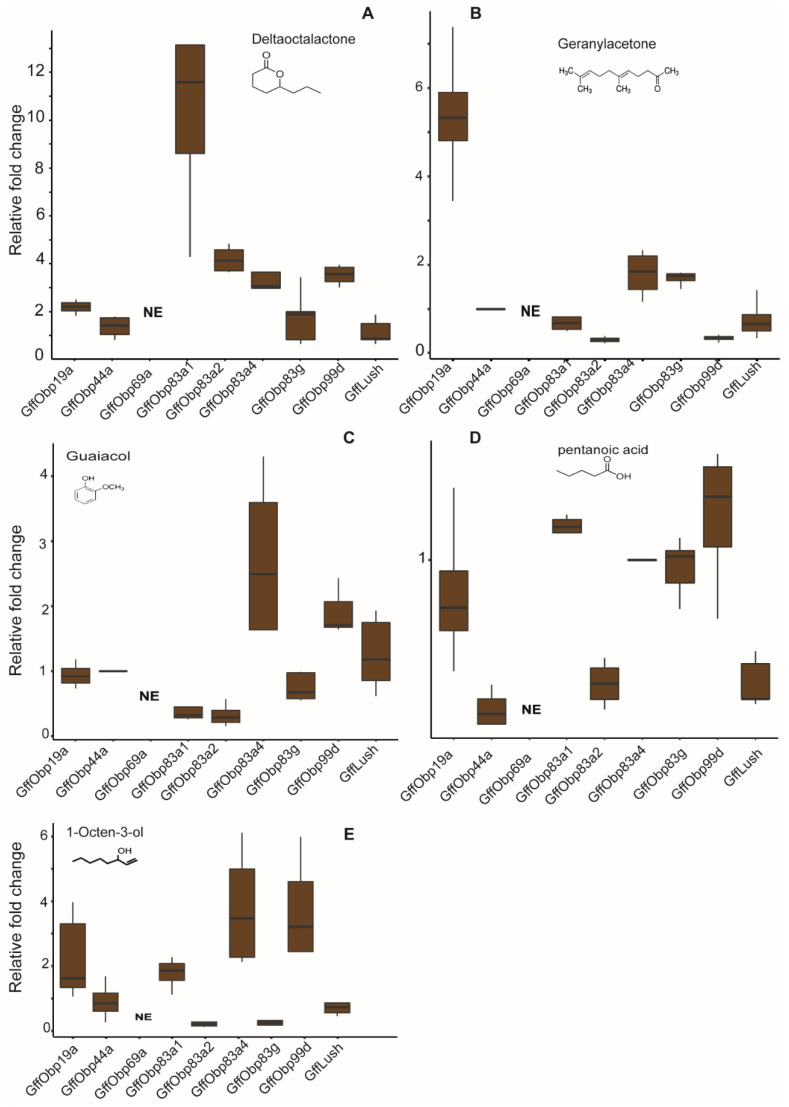
mRNA transcript change patterns when the flies were exposed to different chemicals. (**A**–**E**) Boxplots illustrating the relative fold change when (**A**) flies were exposed to δ-octalactone, (**B**) flies were exposed to geranyl acetone, (**C**) flies were exposed to guaiacol, (**D**), flies were exposed to pentanoic acid and (**E**) flies were exposed to 1-octen-3-ol. NE means that OBPs were not expressed in the antenna.

## Data Availability

The data used to support the findings of this study are available from the corresponding author upon request.

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
