# Peer review of "Antennal Enriched Odorant Binding Proteins Are Required for Odor Communication in Glossina f. fuscipes"

_biomolecules, 2021, doi:10.3390/biom11040541_

Round 1
Reviewer 1 Report
As I already said in the first round of revision, in my opinion this study has great interest, because it has demonstrated that OBPs are involved in olfaction perception in a vector of human diseases, as sleeping sickness, and could be used to design control tools for this pest. Also, the use of gene silencing techniques, along tissue expression studies and in silico studies of ligan-protein interaction could be a good approach to study olfaction in other non-model organisms of interest.
The authors had applied most of my suggestions in this revised manuscript and I recommend the publication in its present form.
Reviewer 2 Report
The suggested changes have been fulfilled, the manuscript is ready to be accepted for publication.
This manuscript is a resubmission of an earlier submission. The following is a list of the peer review reports and author responses from that submission.
Round 1
Reviewer 1 Report
In this study, the authors investigate the role of Odorant Binding Proteins (OBPs) in the tsetse fly Glossina fuscipes fuscipes olfaction, main vector of sleeping sickness using multidisciplinary approaches. Firstly, they did the structural analysis of 23 OBPs with known sequences in this fly finding 12 classic OBPs, 3 Minus-C OBPs and 8 atypical OBPs. Then, they did an In silico modelling with these OBPs and they selected 9 OBPs that shown more than 30% template similarity. And they study with RT-PCR the gene expression of these 9 OBPs in the antenna and legs of the flies showing that 7 of them could have olfactory function because they are expressed at least in the antenna of one sex. Also, they studied the ligand-OBPs interaction patterns with 5 different odors, 4 that inhibit blood feeding and one that enhance it, using the physiochemical properties and molecular docking of the OBPs they show that different OBPs interact with different odors. Likewise, flies exposed to each of these odorants show different gene expression alterations patterns for each OBPs measured by RT-qPCR. Finally, they have silenced the 4 OBPS that are expressed in the antenna of both male and female feeding flies with the dsRNAi of each OBP and conducted behavioral response assays with the odor that enhance blood feeding (1-octen-3-ol), showing an altered response respect to wild type and control flies in 3 of the 4 silenced OBPs. And Their In silico prediction of the interaction between these OBPs and 1-octen-3-ol were investigated are in line with the behavioral results.
In my opinion this study has great interest, because it has demonstrated that OBPs are involved in olfaction perception in a vector of human diseases, as sleeping sickness, and could be used to design control tools for this pest. Also, the use of gene silencing techniques, along tissue expression studies and in silico studies of ligan-protein interaction could be a good approach to study olfaction in other non-model organisms of interest.
Although the study is well written, correctly design and analyzed and their results and conclusions quite interesting. There are some aspects that should be address before publication:
- In section 2.3 about the odor exposure before the expression studies the authors say “Males and females were exposed in a separate cage to avoid mating, which might interfere with the experiments “, while in section 2.9 they say they did the olfactory behavioral assays with gen silenced flies with male and female together. I think that its more important to avoid mating in the olfactory behavioral assays. Why the authors did not separate by sex also in this experiment?
- In section 3.1 the authors should start with a brief explanation of the experiment whose result are going to be shown. Information as: how they get the sequences, the type of analysis they perform, and why they are doing the experiment could help the reader. Also are the 23 OBPs they analyzed the only ones described in this specie? If not, how have the sequences been selected?
- In section 3.2 an explanation of the experiment is also missing.
- In section 3.3 in line 267 the authors say they are using conventional PCR, and I Think they are meaning conventional Retrotranscriptase PCR (RT-PCR), the same happens in Figure 4 C.
- I think that the Supplementary figure 2 should be included in the article to illustrated better the section 3.5
Minor revisions:
- There are two sections numbered 2.8.
- Line 324 and 482: what is the meaning of WRB?
- Line 386: Elterations should be Alterations.
- Line 464: I don´t think that the Figure.7 is the one the authors are discussing here.
Reviewer 2 Report
Main points
Introduction: Your Introduction is very poor and quite inaccurate. One wonders how it is possible to prepare such a script on tsetse olfaction without letting the reader know what has been established about detection of host odors by the olfactory system in this group of flies. A rich literature exists stretching back to the identification of 1-octen-3-ol as an olfactory stimulant and attractant for tsetse from cattle by Dr. David Hall and colleagues in 1984. There have been many other significant reports relating to perception of volatile metabolites associated with ruminants for tsetse including the pioneering work in Nairobi by Margaret Owaga on G. Pallidipes in 1985 and more recent works. It is very surprising how you have not summarized this for the reader. It is quite difficult to treat the question of possible interactions between host compounds and tsetse OBPs without considering the physicochemical properties of the compounds involved.
Attractant biosaasy: Your tsetse attractant bioassay raises many questions and contains some major statistical errors.
Tsetse are active twice daily during the photophase. Why wait for 24h for a response to a highly attractive host chemostimulant?
What was the distribution of the tsetse in the cage after 24h: numbers on test, numbers on control, numbers elsewhere in the cage, number of flies dead? It is against such a distribution in controls that treatment-fly responses should be measured. This data needs to be tabulated to allow the reader appreciate how the assay worked, with details presented regarding the outcome of each of the three replicates for each treatment. You provide no numbers, yet it is patent you encountered mortality with G. f. fuscipes starved for fours days. Further, you need to provide a measure of the variance in all controls and in all treatment situations to the reader. Viewing the distribution of the data in section E of Figure 4 it is obvious that data sets are not normally distributed and that the variance is not homogeneous, contrary to what you state.
Statistics on attractant bioassay data: Both the Sharpio test for normailty and the Levene test for homogeneity of variance indicate highly significant values, proving that data sets from bioassays with tsetse are not normally distributed, nor do they show the same variance. This can, in any case, be readily seen from an examination of the box plots on Fig. 4E. You are flouting fundamental statistical principles by applying an analysis of variance to such data sets.
Silencing OBPs: You state “We silenced these OBPs using RNAi interference technique…” but you present no substantial evidence in support of this statement. What is presented on Fig. 4C is insufficient (see below under Results)
More specific points
The Introduction is not up to the mark.
- Your statement with regard to OBPs being absent from non-flying insects is incorrect: OBPs have been identified in a range of non-flying insects, staring with primitive bristle tails and firebrats.
- Your allusion to vector control tools is quite incomplete: To provide the reader with a comprehensive view you will need to be more explicit. Excellent visual targets have been developed to control tsetse, particularly for G. f. fiscipes, through considerable experimentation by workers across Africa. Please refer to those studies.
See for example in relation to controlling G. f. fuscipes: Rayaisse, JB., Courtin, F., Mahamat, M.H. et al. Delivering ‘tiny targets’ in a remote region of southern Chad: a cost analysis of tsetse control in the Mandoul sleeping sickness focus. Parasites Vectors 13, 419 (2020). https://doi.org/10.1186/s13071-020-04286-w
- Your statement that tsetse olfaction “is the main sensory modality used to locate its hosts” is probably not true for these diurnal flies. Here again you are not doing credit to extensive publications by workers across Africa all published in the past decade showing host odors have only a marginal effect on the efficacy of visual targets for tsetse. Tsetse constitute an important group of insects and their olfactory system merits investigation as such. Nevertheless, should tsetse olfaction form the basis/hypothesis for your work, then you need to explain how your new insights into tsetse olfaction might contribute to intervention strategies for tsetse control?
- Your statement that “the function of OBPs in odor navigation in insect in general” has not investigated is incorrect: OBP's have been most extensively studied in moths where odor navigation is best understood in insects.
- When stating that odorant receptors “are better functionally characterized in various insects” you should indicate which insects you have in mind.
- Summarize what is known for OBPs and their functions in insect vector of disease.
Materials & Methods
Under "Biological Material"
- What were the lighting conditions used for maintaining the G. f. fuscipes colony?
- Provide the reference for the in-vitro feeding system employed to feed tsetse on blood.
Under “Odorants exposure and RNA extraction”
- What is the hypothesis behind this experiment?
- How were tsetse exposed to vapors of test volatiles? What amounts of volatiles were used?
- How did you remove antennae from flies?
- Provide the address of supplier Thermo Scientific
- Provide reference(s) for the statement that “several studies have demonstrated the abundance of OBPs in” legs of insects.
Under “dsRNAi preparation and its delivery to flies”
- indicate to where “the PCR product was sent for sequencing.”
- dsRNAi: Which ones were delivered in blood to flies?
- Quite an amount of lysis sets in when blood is stored even for short period. How did you control for stability of the dsRNAi in the blood?
Under “Behavioural bioassay with dsRNAi gene silenced flies.”
- What were the temperature, humidity and lighting conditions for the attraction bioassay?
- What was the nature of the glue on sticky papers?
- The cotton roll/stick depicted on Figure 4 has both a length and diameter? You only provide one dimension.
- What indicated to you that a source dose of 100 μg 1-octen-3-ol would be attractive to tsetse in the cage?
- G. f. fuscipes starved for three days will show mortality. You have not accounted for this.
- You state “each experiment was replicated three times”. What constituted an experiment?
- Which treatments were tested repeatedly?
You indicate how the image on Figure 3 is a gel electrophoresis readout, but how this protein separation was achieved is not detailed in Materials and Methods. You also do not detail the number of times the separation was repeated on the different tsetse appendages examined?
Results
Under “Tissue-specific expression of different OBPs” Explain to the reader where the readouts depicted on Figure 3 came from? How many times were such expressions of OBPs in tsetse appendages recorded?
Under “The olfactory function of Glossina f. fuscipes OBPs expressed in the antenna” You state “We silenced these OBPs using RNAi interference technique…” but you present no substantial evidence in support of this statement. What is presented on Fig. 4C is not sufficient due to
- the small size and poor quality of the readouts presented,
- the absence of a readout for the control with nuclease free water in males and females,
- the poor labelling - what does M stand for?
- the absence of more than one band in some treatments - two bands are missing for Obp19a-\-
- no indication is presented to show what you did was repeatable in either tsetse sex.
You have not presented data in Results that convincingly shows you have silenced anything.
- Free flight bioassay: You cannot possibly suggest that a cage 75cm long, 30cm wide and 45cm high represents a free-flight assay cage. G. f. fuscipes had hardly taken off when it encountered a cage wall.
- On lines 295-301 you present P values for attraction indices from the attraction bioassay. Which statistical test allowed you ascribe these P values? Nothing is written about this in Materials and Methods.
- Further, the reader needs to see the data that allowed you establish each attraction index over three replicates. You also need to present the distribution of flies captured on the sticky papers for controls where flies imbibed nuclease-free water in blood?
- You state “Similar results were obtained using a double, triple, and a quadruple silencing (Data not shown).” Why is this extra data not explained and presented?
- Legend to Figure 4C : The amount/quality of information presented is insufficient to warrant conclusions regarding silencing OBPs in G. f. fuscipes. Further, you did not work on “mutant” G. f. fuscipes.
Under “Physiochemical properties and molecular docking of G. f. fuscipes OBPs”
- What is WRB?
- Please explain how WRB compounds inhibit feeding by G. f. fuscipes?
- Lines 329-331 “These OBPs strongly interacted with pentanoic acid, δ-octalactone, geranylacetone and guaiacol” - What do these four substances have in common to permit such an interaction with these two OBPs?
- Surely the way to understand which OBP interacts with host compound is to undertake OBP competitive binding studies and establish dissociation constants.
Under “Conformational Dynamic of Docked Systems”
- You state how “the GffObp19a_1_octen_3_ol system showed the highest level of structural compactness” yet GffObp19a-treated flies are recorded to respond no different to controls in the attraction bioassay (see Figure 4).
Under “Evaluation of Deorphanization of Receptors based on Expression Alterations in mRNA levels (DREAM) on different OBPs”
- You have not explained the hypothesis behind this kind of experiment. The OBPs you list could be upregulated for a number of reasons that have nothing to do with exposure to a particular compound. Where is the data for adequate controls?
- No detail is provided on the use of gel electrophoresis in Materials & Methods.
Discussion
- This section is highly speculative in the absence of OBP competitive binding studies and estimated dissociation constants.
You conclude by stating: “We believe that a better understanding of OBPs in chemical communication will contribute to the more efficient development of olfactory-based tools…….for tsetse fly, a vector of sleeping sickness and nagana.”
Tsetse constitute an important group of vectors of disease in sub-saharan Africa, so they are worth studying in their own right. You make statements in the Introduction and at the end of the Discussion regarding the possible contribution of studies on tsetse olfaction to improvement of intervention strategies for this group of vectors. What you write does not represent the reality of the situation that has emerged in the last decade with regard to tsetse control tools based solely on visually attractive targets for these diurnal flies. It is incumbent on researchers, and particularly those working in the field of insect vectors of disease, to provide an objective, impartial and up-to-date account of the situation.
The text contains many inconsistencies in writing and grammatical errors.
An annotated version of the MS with many corrections and comments accompanies this review.

Reviewer 3 Report
The manuscript brings new and significant information about odorant bind proteins. The work seemed sound and is well presented with clear objectives and application of the necessary methodologies to highlight the facts. The figures are well dimensioned and clear, however, although passable, Fig 1 could be visually improved.
Minor points
Line 72-Specify which institution the "Center" belongs to.
Line 79-What does “dilutes at 10-3 v/v” mean? Make the sentence more understandable
Fig 2- It should bring information on how the structural models of proteins were obtained
Line 428- “Our results suggest” -Specify which results suggest what the authors mention.